# The Role of ADAMTS Proteoglycanases in Thoracic Aortic Disease

**DOI:** 10.3390/ijms241512135

**Published:** 2023-07-28

**Authors:** Marsioleda Kemberi, Yousuf Salmasi, Salvatore Santamaria

**Affiliations:** 1Barts and the London School of Medicine and Dentistry, Queen Mary University of London, London E1 2AD, UK; m.kemberi@smd19.qmul.ac.uk; 2Department of Surgery and Cancer, Imperial College London, London W6 8RF, UK; y.salmasi@imperial.ac.uk; 3Department of Biochemical and Physiological Sciences, School of Biosciences, Faculty of Health and Medical Sciences, Edward Jenner Building, University of Surrey, Guildford GU2 7XH, UK

**Keywords:** thoracic aortic aneurysm and dissection, ADAMTS, proteoglycans, aggrecan, versican

## Abstract

Thoracic aortic aneurysm and dissection (TAAD) are complex disease states with high morbidity and mortality that pose significant challenges to early diagnosis. Patients with an aneurysm are asymptomatic and typically present to the emergency department only after the development of a dissection. The extracellular matrix (ECM) plays a crucial role in regulating the aortic structure and function. The histopathologic hallmark termed medial degeneration is characterised by smooth muscle cell (SMC) loss, the degradation of elastic and collagen fibres and proteoglycan (PG) accumulation. Covalently attached to the protein core of PGs are a number of glycosaminoglycan chains, negatively charged molecules that provide flexibility, compressibility, and viscoelasticity to the aorta. PG pooling in the media can produce discontinuities in the aortic wall leading to increased local stress. The accumulation of PGs is likely due to an imbalance between their synthesis by SMCs and decreased proteolysis by A Disintegrin-like and Metalloproteinase with Thrombospondin motifs (ADAMTS) proteoglycanases in the ECM. Mouse models of TAAD indicated that these proteases exert a crucial, albeit complex and not fully elucidated, role in this disease. This has led to a mounting interest in utilising ADAMTS proteoglycanases as biomarkers of TAAD. In this review, we discuss the role of ADAMTSs in thoracic aortic disease and their potential use in facilitating the clinical diagnosis of TAAD and disease progression.

## 1. Introduction

Aortic aneurysms are permanent dilations of the aortic diameter by more than 50% involving all three layers of the wall [1]. In the thoracic aorta, progressive dilation and weakening can lead to an aortic dissection/rupture due to an acute tear in the aortic intima leading to the propagation of blood flow through a false lumen. According to the Stanford classification, type A aortic dissections, which involve the ascending thoracic aorta, have a 50% mortality if left untreated, and are far more fatal than type B aortic dissections (which do not involve the ascending or arch aorta), usually managed by medical therapy [2].

In a large European population-based study, the incidence of aortic dissection was found to be 2.53/100,000 persons/year [3]. Data from the Global Burden of Disease Study also found that death from aortic aneurysm-related emergencies occurs at a rate of 2.4/100,000 persons, and is the most common cause of death among conditions requiring emergency surgery in high-income countries [4]. The number of people in the United Kingdom who are 75 or older will double over the next 40 years, and it is estimated that the percentage of incident dissection events in the elderly will rise to 57% by 2050 [5]. Despite this, size criteria have a low sensitivity, and with current diagnostic methods, it is difficult for clinicians to predict upcoming dissections in patients with thoracic aortopathies.

The histopathologic hallmark termed medial degeneration is characterised by smooth muscle cell (SMC) loss, the degradation/disorganisation of elastic and collagen fibres and proteoglycan (PG) accumulation (Figure 1) [6]. The net accumulation of PGs in TAAD is presumed to arise from elevated synthesis and reduced proteolytic degradation over a prolonged time period. The proteases responsible for regulating levels of aortic PGs are members of the A Disintegrin-like and Metalloproteinase with Thrombospondin motifs (ADAMTS) family of metalloproteinases [7].

This crucial function of ADAMTS proteoglycanases has been recently highlighted by mouse models of aortopathies as well as by in vitro studies characterising their molecular interactions with PGs [7,8,9,10]. ADAMTS mRNA/protein levels have also been measured in the tissues and plasma/serum of TAAD patients [7].

This literature review covers research articles published from January 1997 to April 2023. A comprehensive search strategy was deployed, encompassing multiple academic databases, including PubMed, Scopus, and Web of Science. The primary search terms utilised were “ADAMTS”, “Proteoglycanases”, “Thoracic Aortic Aneurysms and Dissections”, and “Biomarkers”.

## 2. Thoracic Aortic Aneurysms and Dissections

The aorta is the largest vessel in the vasculature, carrying oxygenated blood away from the heart to the rest of the body. Its diameter in healthy adults does not usually exceed 40 mm [1]. Several factors have been found to influence aortic size by increasing haemodynamic stress on the aortic wall, including age, gender, blood pressure and lifestyle [11,12]. In normal aortas, there is a slow and progressive dilation (0.9 mm in men and 0.7 mm in women for each decade of life) caused by a higher collagen to elastin ratio, which leads to increased stiffness and loss of recoil capacity [11,12].

Thoracic aortic aneurysm development is gradual, painless and up to 95% of patients with aneurysms are asymptomatic [13]. This can lead to an aortic dissection, defined as the separation of the medial and intimal layers of the aortic wall [1].

There is a rapid increase in the risk of dissection and rupture when the aortic diameter exceeds 60 mm [14]. A total of 50% of patients with an acute dissection involving the ascending aorta will die at home or before hospital admission [5]. In England, 2500 acute aortic dissection cases occur each year and the gold standard treatment is surgery [15]. In the last decade, a doubling of the total number of operations performed has been noted [15]. With the improvements in surgical techniques in recent years, there has been a trend toward lower mortality, from 23% to 14.7% [15], but the 30-day fatality rate for TAAD still remains high at 73% [5].

Imaging is the only modality currently used to detect an aneurysm, monitor its size, and determine the timing for surgical repair. Current guidelines rely solely on size criteria: they recommend surgical intervention in patients with maximal aortic diameter >50 mm or >45 mm in patients with connective tissue disease (CTD) and/or a bicuspid aortic valve (BAV) as risk factors [16]. Up to 40% of the dissections occur below 50 mm, thus emphasising that the aortic diameter is not sufficient to predict future dissections [17]. Since aneurysmal progression in the aorta is usually asymptomatic, close follow-up and clinical attention is necessary to monitor these patients, and elective surgery should be offered at the optimum time to prevent the occurrence of aortic dissection.

When a type A aortic dissection does occur, it is a clinical emergency and mortality can reach as high as 50% within 48 h [18]. Patients usually present with intense chest pain, and an array of potential other symptoms, including neurological deficit, collapse, breathlessness, paraplegia and abdominal pain, which make the risk of misdiagnosis of this fatal condition unusually high. In this setting, patients should have emergency surgical replacement of the portion of the aorta affected by the entry tear, which may (or may not) include the additional replacement of an acutely incompetent aortic valve or part (or all) of a dissected aortic arch. The degree of complexity of these operations increases as the number of portions of the aorta that need to be replaced increases. This, in turn, also increases the mortality risk, which ranges anywhere between 10 and 50%.

Due to the life-threatening presentation of TAAD and the urgent need for a diagnostic tool, biomarkers such as D-dimer (DD) and matrix metalloproteinases (MMPs) have been thoroughly investigated. DD, a degradation product of cross-linked fibrin and a well-established biomarker for pulmonary embolism, has a good negative predictive value (97.6%) for acute aortic dissection within the first six hours [19]. However, it has low specificity (46.6%) and overlaps with acute coronary syndrome (ACS) [19]. A total of 80% of misdiagnosed dissection cases are mistaken for ACS, which can lead to worse outcomes if thrombolytic therapy is administered [20]. Furthermore, DD values cannot be used in isolation as patients with thrombosed lumen, a shorter dissection length and who are a younger age can have lower DD values [21]. Thus, the extremely high plasma levels of DD in TAAD may only reflect the extensive inflammatory reactions compared to healthy controls.

MMPs have been detected in both aortic aneurysms and dissections of the ascending aorta. These studies mainly focused on MMP1, MMP2, and MMP9 due to their ability to cleave elastin and collagen, crucial components of the aortic ECM [22,23]. Elevated plasma levels of MMP9 are found following aortic dissection with maximal levels occurring approximately 2 weeks after the event [22]. MMP9 levels did not correlate with aortic diameter changes, although diameter size is not a valid predictor of dissection [22,24]. Moreover, immunostaining of pathological aortas has revealed elevated levels of MMP7, which is responsible for the degradation of numerous ECM proteins [23].

In summary, although MMPs may be implicated in TAAD, it has not yet been established whether they can be regarded as reliable biomarkers for TAAD.

Other candidate biomarkers for TAAD include smooth muscle myosin heavy chain, calponin and soluble elastin fragments [21]. More recently, circulating microRNAs have been recognised to have promising clinical value in diagnosing acute aortic dissections [25]. However, despite studies reporting differential expression of microRNAs between aortic dissection patients and healthy control subjects, the results have been inconsistent, possibly due to the genetic heterogeneity of TAAD, making it difficult to identify a specific and sensitive microRNA as a potential biomarker [26].

Overall, the disappointing performances of these candidate biomarkers have stimulated investigations into alternative molecules whose differential levels in TAAD patients, as compared to non-aortopathic subjects, may be used to monitor disease progression.

## 3. The Extracellular Matrix

Structurally, the aorta is composed of three main layers: the tunica intima, media, and adventitia (Figure 1). The tunica intima, located on the innermost surface of the aorta, consists of a monolayer of endothelial cells supported by a basement membrane [27]. The main role of the intima is to provide a smooth and non-thrombogenic surface for optimal blood flow [27]. The majority (77–80%) of the aortic thickness is occupied by the media, which is primarily composed of SMCs and their surrounding ECM [28]. SMCs are separated by elastic fibres which allow the aorta to expand and contract in response to the pulsatile blood flow generated by the heart [27]. The tunica adventitia is the outermost layer and consists mainly of fibroblasts and collagen fibres. It contains small penetrating blood vessels called vasa vasorum and provides additional support and structural integrity to the aortic wall [27].

SMCs regulate the mechanical properties of the aorta by modulating the ECM composition. This occurs through a finely tuned turn-over of its structural macromolecules, predominantly collagens, elastin and PGs, mediated by proteases belonging to the MMP and ADAMTS families, respectively. This activity is post-translationally regulated by tissue inhibitors of metalloproteinases (TIMPs) [29], as well as by endocytosis mediated by the low-density lipoprotein receptor protein 1 [30,31].

Beyond its major structural role, the ECM regulates cellular communication and function by assisting the transport of cytokines and growth factors between cells [32]. Patients with CTD, including Marfan syndrome (MFS), Ehlers–Danlos and Loeys–Dietz, are at higher risk of developing TAAD due to the widespread defects in the ECM composition via transforming growth factor-beta (TGF-β) signalling pathway upregulation [33]. Enhanced TGF-β signalling is known to promote the fragmentation of elastin and collagen fibres by MMPs and induce the expression of PGs [33]. Only 20% of the TAADs are attributable to CTDs and most of the TAADs are sporadic, i.e., they are not related to an identifiable mutation [34]. However, gene expression studies in these patients have revealed that there may be an alteration in the expression pattern of genes responsible for the maintenance of the ECM [35].

### 3.1. Proteoglycans and Biomechanical Properties of the Aorta

PGs are expressed by both endothelial cells in the intima and SMCs in the media [28]. They comprise a protein core and covalently bound glycosaminoglycan chains (GAGs) composed of negatively charged disaccharide units [28]. The most abundant GAG subtypes in aortic PGs include chondroitin sulphate (CS) (40–62.5%), followed by heparan sulphate (18–30.5%) [36]. GAGs attract cations like Na+ from the interstitial fluid to maintain electroneutrality, hence their ability to exert a Donnan osmotic pressure on the tissue [28]. Due to their overall negative charge, they also bind to positively charged molecules, such as cytokines, chemokines and growth factors, which assist a range of processes including cell signalling, migration, proliferation and apoptosis [28]. The aortic ‘proteoglycanome’ comprises at least 20 PGs, including the large aggregating PGs versican and aggrecan, as well as small leucine-rich PGs such as biglycan and decorin and heparan sulfate PGs [6]. Aggrecan has an order of magnitude more CS than versican and biglycan [37] and therefore has more potential to exert osmotic pressure. Both aggrecan and versican interact with hyaluronan (an unsulphated GAG) and create reversible and compressive structures that smooth out pressure waves in blood vessels [38]. PGs provide flexibility, compressibility and viscoelasticity to the aorta and therefore their accumulation results in increased osmotic pressure and local stress [38,39]. In silico studies have demonstrated that GAGs/PGs pooling in the media can produce discontinuities in the artery wall, which lead to stress concentrations [39]. This can disrupt cell–matrix interactions and delaminate the aortic wall structure compromising the mechanosensing properties and ultimately the structural integrity of the blood vessel [39]. Local flow patterns, aortic wall stresses and wall thickness have all been found to influence aortic wall integrity [40]. The pooling of the PGs in the media has an impact on these properties, and due to our limited understanding, these relationships are still in the experimental stage and have not yet been used in clinical diagnosis.

### 3.2. Proteoglycans in Aortic Disease

In a normal thoracic aorta, the overall PG content increases with age up until 30–40 years, beyond which it exhibits a gradual decline concomitant with an alteration in the GAGs subtypes [36]. An acute dissection is found to occur between 62.3–85.3 years of age [5], excluding CTD patients who have an earlier onset, which may suggest that in TAAD, there is no decrease in the overall PG levels after a certain age but, instead, an ongoing accumulation. To support this, Cikach et al. showed that aggrecan accumulation was consistently associated with aortic dissection/rupture in a mouse model of MFS [6].

In the vasculature, versican plays a role in a number of processes, including cell adhesion, proliferation and migration [28]. Five isoforms (V0–V4) arise by the alternative splicing of the central, GAG-rich region, consisting of two subdomains called αGAG and βGAG, both present in the canonical isoform V0 [28]. V0, V1, V2 and V3 have all been shown to be secreted in blood vessel walls, in particular by SMCs [41,42]. Expression of versican is essential for normal development of heart and blood vessels and its levels increase in aneurysmal lesions and atherosclerotic plaques [7].

Biglycan, a small leucine-rich PG involved in the assembly of collagen fibrils [43], is associated with early onset TAAD in humans after a loss of function mutation [44]. Approximately 50% of *bgn* male knockout mice died of spontaneous thoracic (82%) or abdominal (18%) aortic dissections due to biglycan deficiency contributing to the breakdown of collagen and elastin fibres which affects the tension forces in the vessel wall [45].

## 4. The Role of ADAMTS Proteoglycanases in Aortic Disease

In humans, the ADAMTS family comprises 19 secreted members. Six of these have been shown to have proteoglycanase activity in vitro: ADAMTS1, 4, 5, 9, 15 and 20 [7]. Quantitative studies using purified recombinant proteases and full-length versican as a substrate established that ADAMTS5 is the most potent versicanase in vitro, followed by ADAMTS4 and ADAMTS1 [46,47], while data for ADAMTS9, 15 and 20 are not currently available.

For ADAMTS1, 4, 5 and 9, evidence for a role in aortic development and pathology has emerged predominantly from the phenotype of their respective mouse knockouts models (Figure 2). Additionally, although ADAMTS19 proteoglycanase activity has not been characterized in vitro, an involvement in PG regulation has been proposed based on the phenotype of *Adamts19* knockout mice as well as the clinical presentation of patients with aortic valve disease [48,49].

Similar to Fbn1^C1039G/+^ mice (a model of MFS), heterozygous *Adamts1^+/−^* mice when subjected to Angiotensin II (AngII) treatment showed an increased incidence of aortic events compared to wild-type mice [50]. Homozygous *Adamts1* knockout were not investigated in this model, due to elevated perinatal mortality associated with congenital kidney anomalies [51]. Increased nitric oxide (NO) levels in the aortic wall of *Adamts1^+/−^* mice induced MMP9-dependent elastin fragmentation, aberrant collagen deposition and PG accumulation, three hallmarks of medial degeneration [50]. The pharmacological inhibition of nitric oxide synthase (NOS2) reversed aortic dilation and protected *Adamts1^+/−^* mice from aortic pathology [50]. Importantly, patients with MFS showed elevated NOS2 and decreased ADAMTS1 protein levels in their aorta in keeping with the findings in the mice models [50]. These results suggest that ADAMTS1 expression is required to maintain aortic wall integrity. Interestingly, this function of ADAMTS1 was independent of the TGF-β pathway since losartan administration was not able to reduce aortic dilation or media degeneration induced by *Adamts1* deficiency [50].

Another model, where β-aminopropionitrile (BAPN), an inhibitor of lysyl oxidase, (the enzyme responsible for cross-linking collagens and elastin) was used to induce TAAD, gave surprisingly different outcomes. When *Adamts1* expression was abrogated post-natally by tamoxifen injection, TAAD incidence and rupture rates were significantly lower than those in non-induced mice (45.5 versus 81.8% and 18.2 versus 42.4%, respectively) [52]. Medial degeneration and neutrophil/macrophage infiltration were less severe in the *Adamts1^−/−^* mice than in the controls. Decreased inflammation was attributed to the impaired migratory ability of macrophages (in wild-type mice, *Adamts1* expression was found increased predominantly in the adventitia, the major source of inflammatory cells in the aortic wall) [52].

In mice subjected to AngII treatment and a high fat diet, expression of both *Adamts1* and *Adamts4* increased in aortic SMCs and macrophages [53]. *Adamts4* knockout significantly reduced the incidence of AngII/high fat-induced aortic diameter enlargement, aneurysm formation, dissection and aortic rupture [53]. This was reflected in the amelioration of aortic media degeneration, with reduced versican and elastic fibre degradation, macrophage infiltration and apoptosis. However, it is difficult to conciliate these findings with the well-known PG accumulation observed in TAAD patients, unless the detrimental function of ADAMTS4 is not directly linked to its proteoglycanase activity. That this is the case is suggested by decreased SMC apoptosis observed in the aortas of *Adamts4^−/−^* mice and in human aortic SMCs transfected with *ADAMTS4* siRNA, thus pointing to a pro-apoptotic role of this protease [53]. Moreover, ADAMTS4 is present in the atherosclerotic and macrophage-rich areas of the aorta and may be involved in the inflammatory processes of the aortic wall which further degrade ECM components [7].

Although known for its role in osteoarthritis pathology [54], ADAMTS5 is also implicated in maintaining normal versican levels, a function important during cardiac valve development and trabeculation [7]. Similar to *Adamts1^+/−^* mice [50], *Adamts5^−/−^* mice showed increased aortic dilation in the AgII model, associated with the increased expression of versican and reduced versican proteolysis [55]. In *Adamts5^−/−^* mice, increased ADAMTS1 levels had little effect on versican cleavage [7], thus supporting the in vitro data showing that ADAMTS1 has a 1000-fold lower versicanase activity than ADAMTS5 [47]. During development, *Adamts5^−/−^* mice display a range of ascending aortic anomalies such as SMC loss, cell infiltration and increased aortic thickness, predominantly as a result of aggrecan accumulation [56]. These are also observed in *Adamts9^+/−^* mice with high penetrance (73%) [57], suggesting that ADAMTS5 and ADAMTS9 cooperate to maintain a functional ECM during aortic development.

Thus far, genome-wide association studies have not associated variants in *ADAMTS1*, *ADAMTS4*, *ADAMTS5* or *ADAMTS9* with TAAD. Whole exome sequencing in two unrelated families with consanguineous parents identified four individuals with early onset aortic valve disease that carried two rare homozygous loss-of-function *mutations* in *ADAMTS19* [49]. This prompted the generation of *Adamts19* knockout mice to investigate the role of ADAMTS19 and confirm causality. These mice were viable and fertile but developed progressive aortic valve disease characterized by regurgitation and/or aortic stenosis with 38% penetrance, associated with increased cellularity, PG deposition and ECM disorganization in the valves [49]. As mentioned above, thus far, ADAMTS19 has not been characterized for its proteoglycanase activity in vitro, and therefore it remains to be ascertained if the observed PG accumulation observed in the aortic valves was directly related to the loss of *Adamts19*. Aortic anomalies were not described in these *Adamts19^−/−^* mice. Furthermore, three novel loss of function *ADAMTS19* variants were identified in six individuals from three families who were affected by heart valve disease [48]. These patients showed a mild dilation in the ascending aorta, in addition to anomalies in their aortic and pulmonary valves.

BAV is a prevalent congenital cardiac anomaly that predisposes individuals to thoracic aortic aneurysm [58]. It occurs in approximately 2% of the general population, with males representing the majority, accounting for approximately 70% of BAV cases [58]. Notably, Turner syndrome, a genetic disorder characterised by the absence or partial loss of one X chromosome in females, significantly increases the incidence of BAV by at least 50-fold [59]. This observation suggests that the lack of a second X chromosome predisposes both males and females with Turner syndrome to develop both BAV and TAA, collectively referred to as BAV aortopathy [59].

Exome analysis of two cohorts of Turner syndrome patients revealed the presence of risk alleles for *TIMP3*, the endogenous inhibitor of ADAMTS1, ADAMTS4 and ADAMTS5, associated with low expression levels [60]. The hemizygosity (presence of only one copy) of *TIMP1*, resulting from the absence of a complete second X chromosome, synergistically amplified the risk of BAV and aortopathy, by upregulating the TIMP1 targets MMP2 and MMP9, two MMPs endowed with elastolytic and gelatinase activity [60]. The consequent imbalance in ECM breakdown due to elevated MMP/TIMP ratios leads to the structural weakening of the aortic wall [60]. A higher penetrance (75%) of BAV was also observed in *Adamts5^−/−^; Smad2^+/−^* mice [61]. Mechanistically, the lack of ADAMTS5 activity has been shown to interfere with TGF-β signalling through decreased Smad2 phosphorylation. It is not clear how to conciliate these findings with the increased susceptibility to BAV observed in Turner patients. In this context, it will be interesting to cross *Timp3^−/−^* with *Adamts5^−/−^* mice to assess the effect of a combined loss of these interactors on BAV penetrance and severity.

A caveat is that the mouse models used in these studies do not fully mirror human TAAD, as the mice present with a mixture of thoracic and abdominal aortic aneurysms and dissections [53,62]. On this regard, the BAPN model may resemble human pathology more closely, since dissections are mainly located in the thoracic aorta [63].

Levels of ADAMTS proteoglycanases in TAAD patients may shed some light on their role in TAAD development.

## 5. Levels of ADAMTS Proteases in TAAD

Numerous studies have investigated the expression levels of ADAMTS proteoglycanases in TAAD patients, although these must be replicated in larger cohorts and different populations (Table 1). An additional limitation is that different techniques were used to measure protein levels in tissue (immunoblot, which as a technique is intrinsically qualitative and/or semi-quantitative at best) and in plasma/serum (ELISA), thus making it difficult to compare different studies. The quality of the antibodies used for detection may be a cause of concern in the absence of an extensive validation; different antibodies may lead to different results in different studies. The epitope recognised by these antibodies is generally not reported, nor is it known if complex formation with other blood proteins affects antibody detection. Finally, plasma levels may not directly correlate with tissue levels since ADAMTS proteases are ECM-bound. Taking altogether these limitations, in this section we will try to draw some conclusions based on the current literature.

From a clinical perspective, studies investigating serum/plasma levels are more relevant than those analysing aortic tissues.

Among the three major proteoglycanases, ADAMTS5 protein has been found to be decreased in plasma [71] and increased in TAAD tissue compared to non-TAAD controls [66]; its mRNA levels were found to be decreased in two studies [6,70]. In a study which included 83 patients, an independent association was established between decreased plasma levels of ADAMTS5 and an increased risk of acute aortic dissection [71]. Notwithstanding the limitations discussed above, it seems likely that ADAMTS5 protein levels are decreased in TAAD patients, raising the intriguing possibility that the increased PG levels characteristic of medial degeneration are, at least partially, due to decreased PG clearance.

Three studies found increased ADAMTS4 levels in aortic tissues from TAAD patients [53,68,69], while one study also found higher ADAMTS4 levels in the serum of type A acute dissections compared to controls, with a good diagnostic value (sensitivity: 94.59%; specificity: 97.06%) [69]. However, ADAMTS4 is mainly produced by macrophages and its levels also increase during atherosclerotic lesion development [7], hence inflammation may represent a confounding factor. ADAMTS4 mRNA levels were found to be either increased [53] or unchanged compared to controls [6]. A recent study by Kaufmann et al. identified a low-molecular-weight cyclic peptide with a high selectivity for ADAMTS4 which acted as a molecular magnetic resonance imaging (MRI) probe. The signal strength enabled the prediction of AAA expansion in a mouse model [72]. Although it will be very useful to extend these studies to TAAD, this is an important step in visualising and quantifying targets of ECM proteins in a non-invasive manner which helps increasing our understanding of the pathophysiological processes behind the disease.

Four studies found increased ADAMTS1 protein levels in the tissues of TAAD patients [65,66,67,68], one study found decreased protein levels in the tissues of MFS patients [50] and two studies found increased plasma/serum levels compared to controls [65,69]. One study found higher ADAMTS1 levels in the serum of type A acute dissections compared to the controls, with a good diagnostic value (sensitivity: 87.84%; specificity: 97.06%) [69]. Since a quantitative study has recently shown that ADAMTS1 and 4 proteoglycanase activities are significantly lower than ADAMTS5 [47], and it is possible that such high ADAMTS1 levels may represent a compensative response by SMCs, possibly elicited by PG accumulation. Among the other proteoglycanases, only ADAMTS9 and ADAMTS20 mRNA levels were investigated in the tissues of TAAD patients, and these were found to be unchanged [6].

## 6. Conclusions and Future Perspectives

ADAMTS proteases play a crucial role in regulating the structure and function of the ECM by cleaving PGs. However, the observation that loss of individual *Adamts* genes in TAAD models led to either the exacerbation of or protection from aortic events draws a more complex picture where alternative substrates may be at least as relevant as PGs in determining biological functions. For example, ADAMTS4 may be involved in promoting SMC apoptosis, another feature of media degeneration, through its ability to translocate in the nucleus and there cleave poly ADP ribose polymerase-1 (PARP-1), a key enzyme necessary for DNA repair and genomic stability [53].

Despite advances in understanding ADAMTS proteoglycanases’ involvement in TAAD, significant gaps in knowledge persist. Differences in the choice of the model (BAPN versus AngII, for example) and time points selected for the abrogation of *Adamts* function (embryonic or post-natal) make it even harder to assess their contribution to TAAD. The substrate repertoire of the ADAMTS proteases discussed in this review extends beyond PGs [7,73]. Additionally, it cannot be excluded that the ADAMTS proteoglycanases discussed in this review may exert functions that are independent of their proteolytic activity. Proteolytically independent biological functions have been reported for ADAMTS1 [74], 4 [75] and 5 [76], although their relevance for aortic disease is unknown.

Moving forward, comprehensive mechanistic and preclinical studies are necessary to assess their potential application as pharmaceutical targets or diagnostic biomarkers for TAAD. The identification of additional substrates and proteolytically independent functions should be a priority in future research. Furthermore, studies should replicate findings in larger and diverse patient cohorts to validate ADAMTSs as potential biomarkers for TAAD. Moreover, the development of non-invasive imaging techniques, such as molecular MRI probes, holds promise for visualizing and quantifying ECM protein targets in vivo, providing further insights into the disease’s processes.

In conclusion, the roles of ADAMTS proteoglycanases in TAAD pathology are intricate and multifaceted. At the present time, it is premature to define individual proteoglycanases as targets/anti-targets in TAAD and/or as biomarkers. A deeper understanding of their biological functions, substrate specificity and potential therapeutic applications can provide valuable insights into their pathophysiological role. Due to the lack of therapeutic strategies and biomarkers, ADAMTS proteoglycanases represent a fertile line of research that may deliver a breakthrough in the way that clinicians diagnose and treat this elusive, lethal disease.

## Figures and Tables

**Figure 1 ijms-24-12135-f001:**
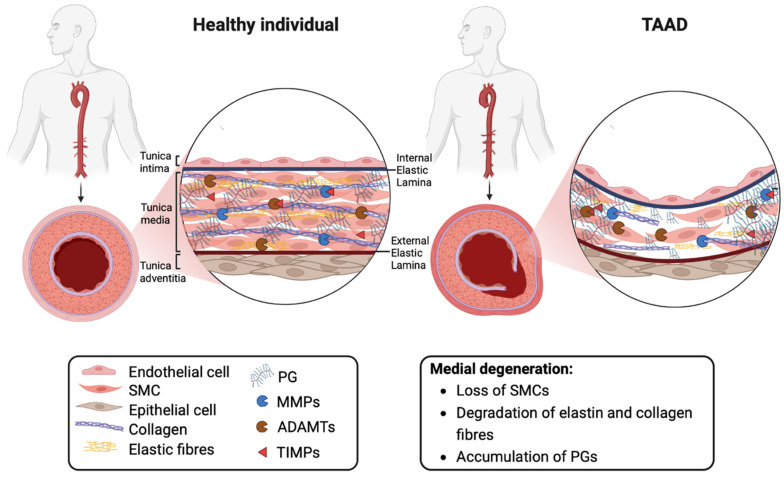
Extracellular matrix remodelling in the normal aortic wall and in thoracic aortic aneurysms and dissections. Abbreviations: ADAMTS, A Disintegrin-like and Metalloproteinase with Thrombospondin motifs; MMP, matrix metalloproteinase; PG, proteoglycan; SMC, smooth muscle cell; TIMP, tissue inhibitor of metalloproteinase. Image created with BioRender.

**Figure 2 ijms-24-12135-f002:**
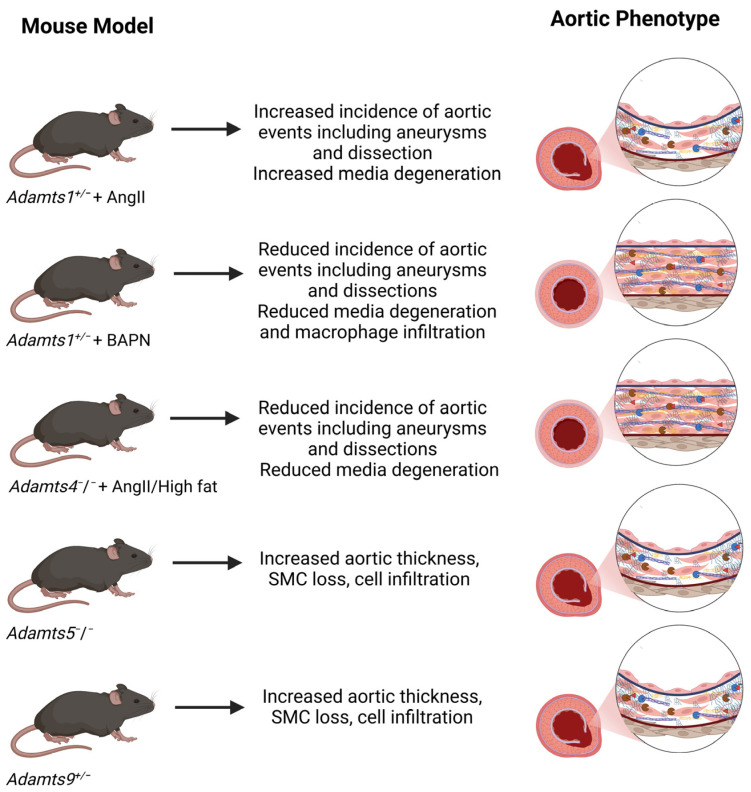
Aortic phenotype of *Adamts* knockout mice. Abbreviations: AngII, angiotensin II; BAPN, β-aminopropionitrile. See text for references.

**Table 1 ijms-24-12135-t001:** Summary of studies investigating ADAMTS proteoglycanases in TAAD patients.

Enzyme	Disease	Expression	Level	Localization	Reference
ADAMTS1	TAAD (type B)	Decreased	mRNA	Tissue	[64]
	MFS	Decreased	protein	Tissue	[50]
	TAAD	Increased	protein	Plasma/Tissue	[65]
	TAAD	Increased	protein	Tissue	[66]
	TAAD	Unchanged	mRNA	Tissue	[6]
	TAAD	Increased	mRNA	Tissue	[67]
	TAAD	Increased	mRNA/protein	Tissue	[68]
	TAAD (type A)	Increased	mRNA/protein	Serum/Tissue	[69]
ADAMTS4	TAAD	Increased	mRNA/protein	Tissue	[68]
	TAAD	Unchanged	mRNA	Tissue	[6]
	TAAD	Increased	protein	Tissue	[53]
	TAAD (type A)	Increased	mRNA/protein	Serum/Tissue	[69]
ADAMTS5	TAAD	Increased	protein	Tissue	[66]
	TAAD	Decreased	mRNA	Tissue	[6]
	TAAD	Decreased	mRNA	Tissue	[70]
	TAAD	Decreased	protein	Plasma	[71]
ADAMTS9	TAAD	Unchanged	mRNA	Tissue	[6]
ADAMTS20	TAAD	Unchanged	mRNA	Tissue	[6]

## Data Availability

Not applicable.

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
