# Peer review of "The Role of ADAMTS Proteoglycanases in Thoracic Aortic Disease"

_ijms, 2023, doi:10.3390/ijms241512135_

Round 1

Reviewer 1 Report

This is a review article, which aims to discuss the role of ADAMTSs in thoracic aortic disease and their potential biomarker for the clinical diagnosis and progression of TAAD.

Generally, the topic is quite interesting and well-presented, and the authors have in depth knowledge. The findings are sufficiently well-presented, clear, and easy to understand, so as to reach safe and solid conclusions. Overall, the manuscript is well written and structured. Thus, I think it would make a nice addition to IJMS.

However, the following points should be generally considered, thus minor revision is demanded.

1)      Line 37; Please provide the precise classification of TAA.

2)   Line 39-41; Kindly present the respective rates that are applicable for the European population.

3)    Although it is a review article, please describe shortly the methods that you used for the literature search.

4)  Line 44; Short description of disease clinical manifestations, diagnostic approaches, as well as therapeutical interventions, mortality and morbidity rates would be a nice addition.

5)      Line 52-54; Please add citation.

6)      Line 62-63; Briefly describe how these factors contribute to the development and progression of the disease.

7)      Manuscript and Figure 2 should be aligned. Please add ADAMTS1 along with ADAMTS4 and AngII/High fat in the figure, otherwise clarify.

8)  Line 318; Please double check the references, they should be with the manuscript, tables, and reference list.

9)      Conclusion needs to be short and clear, highlighting the key message of this review.

10)   Consider modifying it adding separate paragraphs for limitations and future perspectives.

This is a review article, which aims to discuss the role of ADAMTSs in thoracic aortic disease and their potential biomarker for the clinical diagnosis and progression of TAAD.

Generally, the topic is quite interesting and well-presented, and the authors have in depth knowledge. The findings are sufficiently well-presented, clear, and easy to understand, so as to reach safe and solid conclusions. Overall, the manuscript is well written and structured. Thus, I think it would make a nice addition to IJMS.

However, the following points should be generally considered, thus minor revision is demanded.

1)      Line 37; Please provide the precise classification of TAA.

2)   Line 39-41; Kindly present the respective rates that are applicable for the European population.

3)    Although it is a review article, please describe shortly the methods that you used for the literature search.

4)  Line 44; Short description of disease clinical manifestations, diagnostic approaches, as well as therapeutical interventions, mortality and morbidity rates would be a nice addition.

5)      Line 52-54; Please add citation.

6)  Line 62-63; Briefly describe how these factors contribute to the development and progression of the disease.

7)      Manuscript and Figure 2 should be aligned. Please add ADAMTS1 along with ADAMTS4 and AngII/High fat in the figure, otherwise clarify.

8)  Line 318; Please double check the references, they should be with the manuscript, tables, and reference list.

9)     Conclusion needs to be short and clear, highlighting the key message of this review.

10)  Consider modifying it adding separate paragraphs for limitations and future perspectives.

Reviewer 2 Report

Very interesting and well-writed paper. The concerns about this field are well organized so I believe that no suggestions needed

Author Response

We thank the reviewer for the positive feedback.

Reviewer 3 Report

This review about ADAMTS proteoglycanases role in TAAD is globally well written and organized. 

I have few minor suggestions for improving the quality of this. work before considering it fully acceptable for publication:

1)    In paragraph 4 consider to implement the. description by adding information from the following refs: 

a.     doi: 10.1161/CIRCRESAHA.118.313737.

b.     doi: 10.3389/fmolb.2021.701959.

c.      doi: 10.1159/000521498.

2)    Please, add a Table with in vitro cell types, cell treatments, and/or cell model used, the studied ADAMTS and the results obtained from them

3)    In paragraph 5 please add a  descriptive statement (or a Table if better) about the circulating forms (soluble or not) of the different ADAMTS and the eventual carriers

4)    In paragraph5, lines 331-343, please consider to add information from the following ref: https://doi.org/10.3390/biom12010012,
